# Effectiveness, acceptability, and feasibility of technology-enabled health interventions for adolescents living with HIV in low- and middle-income countries: A systematic review protocol

Talitha Crowley[1][*], Charne Petinger[2], Brian van Wyk[2]

**1** School of Nursing, Faculty of Community and Health Sciences, University of the Western Cape, Cape Town, South Africa, **2** School of Public Health, Faculty of Community and Health Sciences, University of the Western Cape, Cape Town, South Africa

⊕ These authors contributed equally to this work.
* tcrowley@uwc.ac.za

**Data Availability Statement:** No datasets were generated or analysed during the current study. All

## Abstract

Adolescents living with chronic conditions such as HIV (ALHIV) are challenged to remain adherent and engaged in HIV care. Technology offers a promising platform to deliver behaviour-change interventions to adolescents. The largest proportion of ALHIV resides in sub-Saharan Africa; yet little is known about the effectiveness, feasibility and acceptability of technology-enabled interventions to deliver and support health care to ALHIV in resource-constraint settings. This study aims to explore the literature and synthesise the evidence for the effectiveness, acceptability, and feasibility of technology-enabled health interventions for ALHIV in low and middle-income countries (LMIC). Eight electronic databases (Ebsco-host, CINAHL, ERIC, MEDLINE, PubMed, SCOPUS, Science Direct, and Sabinet) and Google Scholar will be searched to identify technology-enabled health interventions for ALHIV in LMIC published from 2010–2022. Quantitative and qualitative studies reporting on technology-enabled health interventions for predominantly adolescents (10–19 years) will be included. The review will be performed, and findings reported according to the Preferred Reporting Items for Systematic Reviews and Meta-analyses Protocols. A two-stage process of screening titles and abstracts, and then full-text, will be performed independently by two reviewers. The quality of the included studies will be assessed using the Critical Appraisal Skills Programme checklists, and the Risk of Bias in Non-randomised Studies of Interventions tool will be used to assess the risk of bias. The review will involve publications already in the public domain; therefore, ethics approval is not required. The results will be disseminated through a peer-reviewed journal publication and/or conference proceedings.

**PROSPERO registration number:** CRD42022336330.

relevant data from this study will be made available upon study completion.

**Funding:** The author(s) received no specific funding for this work.

**Competing interests:** The authors have declared that no competing interests exist.

## Introduction

Adolescents living with HIV (ALHIV) globally are challenged to consistently adhere to antiretroviral treatment (ART) and achieve and maintain viral suppression. It is reported that adolescents between the ages of 10–19 years have lower adherence and retention in care compared to adults (over 20 years) and children (under 10 years) [1–4]. Whereas self-management of a chronic condition such as HIV is generally difficult, adolescents find themselves in a challenging development life stage where they transition from childhood to adulthood, amidst a myriad of physiological, emotional/psychological and social changes that take place within and around them [5]. ALHIV must take over the responsibility of their health care, in terms of their medication adherence, clinic visits, and relations with health care providers as well as significant adults in society (e.g., school, work and community) [6]. It is recommended that ALHIV be taught a combination of self-management strategies to improve adherence and engagement in care as well as general physical and mental well-being [7, 8].

UNAIDS has identified ALHIV as a key population that should be targeted for specific intervention to fight the global HIV pandemic. Whereas there are approximately 1.75 million ALHIV globally; 80% of them reside in sub-Saharan Africa in 2020 [9]. Living with HIV in lower- and middle-income countries (LMIC) bring about many challenges, such as HIV-related stigma, exposure to stressful life events, poverty, and limited access to adolescent-friendly health care services. However, evidence for behavioural interventions for ALHIV in LMIC is limited, because of the paucity of high-quality intervention studies [10, 11]. Technology-enabled interventions may hold potential as an instrument to deliver intervention modalities to ALHIV in LMIC, as shown in examples in high-income countries such as the USA [12].

Technology plays an integral role in the lives of many adolescents and 82.8% of youth in the age range of 11 to 18 years spend on average 1 to 4 hours per day online [13]. For most adolescents, the internet is their preferred source of health information [13, 14]. Technology-enabled health interventions (also referred to as technology-delivered or simply, technology-based) use electronic devices such as mobile phones or computers for health information communication [15, 16]. These interventions are accessed through a device that can be connected to a mobile network or the internet in the form of an app on a mobile device or on the World Wide Web [16]. Recently there has been an increase in technology-enabled health interventions for adolescents [17]. Technology platforms enable the transfer of health information, provider/peer communication and support and can reinforce self-management behaviours through self-assessment, goal setting and problem-solving [17, 18]. Moreover, technological approaches can be convenient as it is private and provides a sense of autonomy for adolescents [5]. Smartphone apps, social networking, virtual reality, and gamification are particularly relevant in the context of HIV to facilitate engagement, privacy, support, and feedback [14].

Several systematic reviews have indicated that technology-enabled interventions can be effective for health promotion [17, 19–22], prevention [23, 24], the management of mental health [25, 26] and chronic conditions in adolescents [12, 16]. Although technology-enabled interventions have increased rapidly and can be used in varied contexts and across different health conditions, their applicability to marginalised groups is limited [27]. Five reviews have found positive effects of technology-enabled health interventions on the health and treatment outcomes of adults living with HIV. Areri et al. [7] found that technologically assisted interventions for adults living with HIV improved symptom management, quality of life, adherence, and mobilising social support. Technology integrated into standard care improves care access and strengthens the relationship between patients and health care services [28]. Zang and Li [29] found that technology platforms enabled HIV self-management, and reduced health care disparities among people living with HIV. Digital health interventions for HIV and

STIs were highly acceptable and feasible; and mHealth (SMS) and internet-based interventions improved a range of outcomes, including ART adherence, retention in care, risk reduction behaviours and self-care [30]. More recently, Manby et al. [31] reported that e-health interventions for HIV prevention and management in sub-Saharan Africa were a low-cost way to improve HIV management behaviours (adherence and retention in care).

## Rationale

There is a paucity of evidence on the effectiveness of technology-enabled health interventions for ALHIV in LMICs [26, 29]. A previous review on technology-enabled interventions for ALHIV was geographically limited to the USA [12]. The last search date for the global review on HIV, technology, and youth [14] was 2015 and found only one study focused on ALHIV outside the USA. To date, no formal review of technology-enabled interventions for ALHIV in LMIC has been conducted. There has been an increase in the development of technology-enabled interventions globally. A review of interventions (2016–2018) to improve ART adherence and retention in care for ALHIV and youth, found one technology-enabled intervention (mHealth/SMS) [32] and a review on self-management interventions for ALHIV published in 2021 found four technology-enabled interventions and seven ongoing studies with a technology component [33]. Previous reviews have focused mainly on HIV treatment outcomes such as adherence and viral suppression and not on feasibility, acceptability, usability/functionality and perceived usefulness, which is key to understanding why an intervention is effective or not in a particular context and for guiding the development of future interventions. It is necessary to identify design features as well as adolescent preferences for technology-enabled health interventions to guide future intervention development and scale-up. Moreover, there is a need to synthesise the effectiveness, feasibility, and acceptability of various technology-enabled interventions for ALHIV in LMIC.

The current review aims to map the use of technology-enabled health interventions and determine the effectiveness, feasibility, and acceptability of such interventions on health-related outcomes for ALHIV in LMIC.

## Review questions

The review will be guided by the following questions:

1. What technology-enabled interventions have been implemented in LMIC to support and deliver healthcare to ALHIV (aged 10–19 years)?

2. What is the effectiveness of various technology-enabled health interventions on general health and well-being and treatment outcomes of ALHIV in LMIC?

3. What is the feasibility, acceptability, usability/functionality and perceived usefulness of the various technology-enabled health interventions for ALHIV in LMIC?

# Materials and methods

## Study design

The review will be guided by the seven systematic review steps as described by Egger, Davey and Smith [34]. The steps are to formulate the review question, determine the inclusion and exclusion criteria, develop the search criteria, perform the study selection, assess the quality of the studies, extract the data, and analyse and synthesise the data. Amendments to this protocol

will be reported in the published review. The protocol was written according to the guidelines provided in the Preferred Reporting Items for Systematic Reviews and Meta-analyses Protocols (PRISMA-P) (see S1 Checklist) and registered with the International Prospective Register of Systematic Reviews (PROSPERO) on 13 June 2022 CRD42022336330.

## Inclusion and exclusion criteria

We will consider studies to be eligible for inclusion in the review if they meet the following eligibility criteria.

1. Types of participants: Adolescents living with HIV between the ages of 10–19 years. Included studies should include adolescents aged between 10–19 years as the primary study population or report this age group as a sub-analysis. We will also consider an age range of up to 24 years of age as it is difficult to find data specific to adolescents aged 10–19 years. It is postulated that the age range 10–24 years relates better to adolescent growth and development [35].

2. Types of interventions: Studies that describe a technology-enabled intervention to deliver or support healthcare (defined as interventions that use electronic devices such as mobile phones or computers for health information communication).

3. Types of studies: Quantitative (randomised controlled trials, non-randomised controlled trials, before- and after studies) and qualitative studies reporting on the feasibility and acceptability of technology-enabled interventions. Peer-reviewed studies or grey literature will be considered. Studies published in the English language, conducted in LMIC, and published between 2010 and 2022.

4. Types of comparisons: Technology-enabled health intervention vs no intervention, the standard of care, waitlist, or another intervention with no technology-enabled component. We will also consider studies with no comparison.

5. Types of outcomes: We will consider studies reporting on any health-related individual outcomes as defined by the study authors. We will report on data related to the feasibility, acceptability, usability/functionality and perceived usefulness of the intervention.

The PICOT (Population, Intervention, Comparison, Outcome and Time) criteria are summarised in Table 1.

The *exclusion* criteria of the review are as follows:

**Table 1. PICOT- criteria for inclusion of studies.**

| | |
|---|---|
| Patient/ Population | Adolescents living with HIV aged 10–19 years<br>• Should be the dominant study population; or provided in disaggregated sub-analysis |
| Intervention | Any technology-enabled health intervention that delivers or supports health care and is delivered to ALHIV as direct recipients. |
| Comparisons | Not applicable |
| Outcomes | Primary outcomes: health-related individual outcomes as specified by each study e.g., health/ risk behaviours, self-management behaviours, self-efficacy, adherence, retention in care, viral suppression, quality of life, mental health or well-being<br>Secondary outcomes: process outcomes e.g., acceptability, feasibility, usability/functionality, perceived usefulness |
| Time | 2010–2022 |
| Other considerations | English language<br>Low- and middle-income countries as specified by the Organisation for Economic Co-operation and Development [36] |

1. Review studies.

2. Technology-enabled interventions that do not involve the adolescent directly as a recipient of the intervention, i.e., electronic health registers, monitoring and recording of service delivery.

## Information and search strategy

The search strategy will be broad to include technology-enabled health interventions in LMIC for adolescents. An information specialist will be consulted to develop the search strings. The systematic search of databases will be conducted on the following databases: Ebscohost (Psycharticles, Academic Search Premier), Cumulative Index of Nursing and Allied Health Literature (CINAHL), Educational Resource Information Center (ERIC), Medical Literature Analysis Retrieval System Online (MEDLINE), PubMed, SCOPUS, Science Direct, and Sabinet. The full-text articles will be sourced by using the "AND" and "OR" Boolean operators and the following search terms / key words with their MeSH terms. Table 2 contains the search strategy for PubMed. This strategy will be adapted to the syntax and subject headings of other databases, and we will report the full strategy for each database in the final review.

We will search ClinicalTrials.gov (www.ClinicalTrials.gov) and the World Health Organization (WHO) trials portal (www.who.int/ictrp/en/) to identify unpublished and ongoing studies. In addition, we will search grey literature such as university theses/dissertation databases and conference abstracts, for example, the International AIDS Conference, the Conference on Retroviruses and Opportunistic Infections (CROI) and the International Workshop on HIV and Adolescence. In addition, a search will be done on Google Scholar. To complement the electronic search, we will also screen reference lists of included studies and relevant systematic reviews. Specialists in the field and authors of the included studies will be contacted to identify possible unpublished studies.

## Study selection

The studies will be selected based on the PICOT-mnemonics, as seen in Table 1. The inclusion criteria and search strategy will be utilised for the database search. The selection, reviewing, and reporting of the findings of this review will be done in accordance with the guidelines provided by Preferred Reporting Items for Systematic Reviews and Meta-Analyses (PRISMA) [37]. The number of hits from each database will be recorded and the citations will be imported into Covidence software. Covidence software will allow removing duplicates and recording the number of citations. Two reviewers will screen titles and abstracts, for eligibility for inclusion. For inclusion, the full-text articles of eligible studies will be retrieved and independently reviewed by the two reviewers. Discrepancies will be resolved through discussion, and if needed resolved by a third reviewer. Authors of studies will be contacted in case of missing information. Reasons for excluding studies will be provided. We will depict the process of study selection in a PRISMA flow diagram.

**Table 2. PubMed search strategy.**

| |
|---|
| ("adolescent" OR "young people" OR "teen" OR "teenager") AND |
| ("Information and Communications Technology" OR "ICT" OR "Technology" OR "Technology Enabled" OR "Technology based" OR "gaming" OR "social media" OR "eHealth" OR "mHealth" OR "WhatsApp" OR "SMS" OR "mobile" OR "internet" OR "text message" OR "telemedicine") AND |
| ("HIV" OR "AIDS") AND |
| ("Low-income countries" OR "Middle-income countries") |

## Data extraction

Two authors will independently extract data in Covidence using a pre-specified data extraction form (S1 File). Before starting data extraction, we will pilot the form on one study identified for inclusion and modify it if needed. We will extract data on the study design, characteristics of participants, type of intervention, description of the intervention, outcomes, and setting. A description of the components of the technology-enabled health interventions will be extracted using an adapted form following the 12-item Template for Intervention Description and Replication (TIDier) checklist [38]. This will assist to record important aspects of the intervention such as the name of the intervention, rationale, theoretical foundation, duration and intensity, the type of device, technology design, delivery platform/mode, type of information, the person (s) delivering the intervention and their training, the setting and procedures followed. We will resolve disagreements through discussion or by consulting a third person.

## Risk of bias and quality assessment

The quality of the included studies will be assessed using the Critical Appraisal Skills Programme checklists [39], and the Risk of Bias in Non-randomised Studies of Interventions [40] tool will be used to assess the risk of bias. The risk of bias will be assessed at the study and outcome levels. The two reviewers will independently assess the characteristics of each study, then results will be compared, and differences will be discussed among reviewers.

## Data synthesis

The results of this study will be presented in accordance with PRISMA guidance. One author will enter data extracted from individual studies into Review Manager or Stata Statistical Software v17 for analysis and a second author will check the data entry. We will report risk ratios or odds ratios with 95% confidence intervals for dichotomous data to summarise effects. Continuous data will be presented as the mean difference (similar measurements) or standardised mean difference (for different measurement methods) and standard deviations with 95% confidence intervals.

In the case of missing data, we will contact the study authors to obtain the data and will send a reminder if no response is received. Data will be reported as missing if the authors do not respond after the reminder. If we consider data to be missing at random, we will use only the available data in the analysis.

We expect high levels of heterogeneity. Clinical heterogeneity linked to the participants, intervention, setting, outcome measurement and study design will be described in table format. If we decide that studies are sufficiently homogenous to perform a meta-analysis, statistical heterogeneity will be assessed using $Tau^2$ and $Chi^2$ statistics. We will consider heterogeneity to be significant if $Tau^2$ is more than one, or if the p-value of the $Chi^2$ test is less than 0.1. We will consider an $I^2$ statistic of more than 30% as substantial heterogeneity. Substantial heterogeneity will be investigated using sub-group analysis, to assess whether heterogeneity is linked to certain characteristics of included studies.

Where ten or more studies are included in a meta-analysis, we will explore possible reporting bias by assessing asymmetry in funnel plots to determine whether studies were selectively reported.

Statistical analysis will be performed using Review Manager or Stata Statistical Software v17. As we anticipate high levels of heterogeneity, we will use random-effects meta-analysis to pool data if we consider interventions and study populations to be sufficiently similar. We will not combine data from RCTs and non-RCTs in a single meta-analysis. If we consider

heterogeneity to be high, we will not pool data, but rather present findings in a narrative synthesis and table format.

We will perform the following sub-group analyses, if possible, on primary outcomes to explore heterogeneity:

- Type of device (e.g., phone, smartphone, computer)

- Technology design (e.g., interactive vs non-interactive; individual vs group)

- Delivery platform (e.g., smartphone application, game, web-based, SMS/WhatsApp, social media such as Facebook)

We plan to carry out sensitivity analyses on primary outcomes to examine the effect of studies with a high risk of selection and attrition bias; to examine the effect of various intracluster correlation coefficients (ICCs) where we have adjusted results for clustering; and to examine the effect of imputed data.

Data on acceptability, feasibility, usability/functionality and perceived usefulness from qualitative studies will be explored using thematic analysis and reported narratively. *Acceptability* is defined as the receptivity to or uptake of technology-enabled health interventions by health worker implementers and adolescent recipients. This data can be reported in the form of descriptive surveys or narratives. *Feasibility* is defined as the perceived convenience in using technology platforms or measures and can be reported in, for example, completion rates or qualitative data [28]. *Usability/functionality and perceived usefulness* could be tracked through online metrics or qualitative data.

## Assessing the certainty of the evidence

We will include a summary of findings table that will include key information about the certainty of the evidence, the magnitude of the effect of the technology-enabled health intervention and the sum of available data on the main outcomes. We will assess the certainty of evidence using Grades of Recommendation, Assessment, Development and Evaluation (GRADE) [41]. For the quantitative outcomes, we will consider downgrading the certainty of evidence by assessing the following domains: study limitations, consistency of effect, imprecision, indirectness and publication bias. We will consider upgrading the certainty of evidence if there is a large effect, where a dose-response is seen, and where all plausible residual confounding would reduce a demonstrated effect or would suggest a spurious effect if no effect was observed [42, 43]. For each outcome, we will describe the certainty of evidence to be very low, low, moderate, or high. We will use GRADEPro software to generate summary of findings tables.

## Ethics

Ethics approval is not needed for this systematic review, as it will not involve participants and will utilise publicly available data. As this is a systematic review protocol no patients or the public were involved in the design or research.

## Discussion

ALHIV is a vulnerable population group with generally poor treatment outcomes compared to adults and children. Differentiated care models are required to address their specific needs, with technology-enabled health care interventions showing promise in high-income countries and in adults living with HIV. However, effectiveness, feasibility, and acceptability need contextual consideration. The findings will be useful for health intervention developers and will

provide much-needed information on how technology-enabled health interventions may be useful for ALHIV in LMIC. It will inform the development of new interventions and scale-up of existing interventions by highlighting which interventions appear effective and for what outcomes as well as practical implementation considerations. The findings will be disseminated through publication in peer-reviewed journals and/or conference proceedings.

## Conclusion

The review will map the range of technology-enabled health interventions for adolescents living with HIV in LMIC and provide current evidence on the effectiveness, feasibility, and acceptability of technology-enabled health interventions for ALHIV in LMIC.

## Supporting information

**S1 Checklist. Reporting checklist for the protocol of a systematic review and meta-analysis.** Based on the PRISMA-P guidelines.
(DOCX)

**S1 File. Pre-specified data extraction form.**
(DOCX)

## Author Contributions

**Conceptualization:** Talitha Crowley, Brian van Wyk.

**Methodology:** Talitha Crowley, Charne Petinger, Brian van Wyk.

**Writing – original draft:** Talitha Crowley, Charne Petinger, Brian van Wyk.

**Writing – review & editing:** Talitha Crowley, Charne Petinger, Brian van Wyk.

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
