## [Decision Letter · Decision Letter 0]

3 Jan 2023

PONE-D-22-18587

Effectiveness, acceptability and feasibility of technology-enabled health interventions for adolescents living with HIV in low- and middle-income countries: a systematic review protocol

PLOS ONE

Dear Dr. Crowley,

Thank you for submitting your manuscript to PLOS ONE. After careful consideration, we feel that it has merit but does not fully meet PLOS ONE’s publication criteria as it currently stands. Therefore, we invite you to submit a revised version of the manuscript that addresses the points raised during the review process.

We look forward to receiving your revised manuscript.

Kind regards,

Muhammad Shahzad Aslam, Ph.D.,M.Phil., Pharm-D

Academic Editor

PLOS ONE

Journal Requirements:

 "The funders had and will not have a role in study design, data collection and analysis, decision to publish, or preparation of the manuscript."

Reviewers' comments:

Reviewer's Responses to Questions

**Comments to the Author**

1. Does the manuscript provide a valid rationale for the proposed study, with clearly identified and justified research questions?

Reviewer #1: Yes

Reviewer #2: Partly

2. Is the protocol technically sound and planned in a manner that will lead to a meaningful outcome and allow testing the stated hypotheses?

Reviewer #1: Yes

Reviewer #2: Partly

3. Is the methodology feasible and described in sufficient detail to allow the work to be replicable?

Reviewer #1: Yes

Reviewer #2: Yes

4. Have the authors described where all data underlying the findings will be made available when the study is complete?

Reviewer #1: Yes

Reviewer #2: Yes

5. Is the manuscript presented in an intelligible fashion and written in standard English?

Reviewer #1: Yes

Reviewer #2: Yes

6. Review Comments to the Author

You may also provide optional suggestions and comments to authors that they might find helpful in planning their study.

Reviewer #1: This is a very well-written protocol study. Please consider the following: 

1-Remove links from the manuscript and move them to the references section (e.g., Table 1, other considerations). 

2-Change Box 1 to a table. 

3-Add a discussion section considering the problem statements, objectives, and intended results of this study.

Reviewer #2: Dear Author,

This is a research addressing a vulnerable population (adolescents living with HIV) in countries were prevalence is among the highest in the world, and for that this research is relevant in my opinion. The manuscript is well written. I think it is relevant to think in advance about how this review can be useful as it’s quite challenging. I suggest some comments that I hope can help you improve your review.

To give you some information about my background, I am an adolescent physician and HIV in adolescents is among my current research interest. I have not done any reviews myself I but have previously conducted an SMS intervention targeting adolescents with chronic condition and I have conducted qualitative individual interviews.

• I would remove the last 2 bullet points in the “Strengths and limitations’ part as it is expected that you use the adequate guidelines.

• About the definition of adolescents

In my understanding, you restrain the research to 10-19 years old adolescents, which can be ok. May I suggest you to consider to widen this age range to 10-24, as suggested by Susan Sawyer and al in their article “The age of adolescents”. Of course it’s a wide age range but it fits better this life phase and might be actually easier to include your articles, as we all know that it’s quite hard to find specific data on adolescents.

• First references

I do agree that you must highlight that this subgroup has lower adherence and retention in care, but the references 1 and 2 you use are in my opinion not adequate because their focus is mental health and not adherence or retention in care.

• Review questions

I find it very relevant to clearly formulate the review questions. To put it in simple words if I am correct: you would like to know what has been done and what works. From my short experience, I am afraid that as you point it later (“We expect high levels of heterogeneity”), you will have trouble performing a meta-analysis and providing narrative synthesis on all interventions because they might vary so much. I suggest that you:

o Formulate right from the beginning that you expect high levels of heterogeneity

o Therefore you will have subgroups (as you say later)

o And that you will try to answer your three research questions for each of the subgroup that you mention later (for example text messages interventions, apps, internet websites etc).

• Qualitative review :

Conducting a review on quantitative AND qualitative data is a very ambitious task. I understand you would like to provide a comprehensive view but I suggest some modifications about the qualitative data.

I have learned that one can perform a narrative review of qualitative studies or perform a systematic review (metasynthesis) where one goes beyond the results of the original studies and produces something that is more than their summary. If I am correct you will perform a narrative review for the qualitative part?

Furthermore, there’s a discrepancy, or something I don’t understand about what you name “qualitative outcomes”, reading those two sentences:

- “Previous reviews have focused mainly on quantitative HIV treatment outcomes and not on feasibility, acceptability, usability/functionality and perceived usefulness, which…”

- “Qualitative data on acceptability, feasibility, usability/functionality and perceived usefulness will be synthesized and reported narratively”

It is not really clear for me what is a qualitative outcome, and from my perspective qualitative data help us understand why an individual or a group think an intervention is useful, and therefore can help building future interventions. I would suggest not to select qualitative outcomes on feasibility, acceptability and so on but rather report patients’ perspectives on the intervention they are questioned about.

• There could be 3 questions for each subgroup:

o What exists?

o What works?

o Why does it work or not from adolescents’ perspective? => qualitative data

Of course it’s really simplified, but I leave it to you to see if it can fit the guidelines for systematic reviews and metaanalysis and I wish you good luck.

7. PLOS authors have the option to publish the peer review history of their article (what does this mean?). If published, this will include your full peer review and any attached files.

Reviewer #1: No

Reviewer #2: No

---

## [Author Response · Author response to Decision Letter 0]

13 Jan 2023

Dear Editor,

We thank the reviewers for their time and effort in improving the manuscript. We have addressed the reviewer comments in a separate document in table format.

Kind regards

Talitha

---

## [Decision Letter · Decision Letter 1]

30 Jan 2023

PONE-D-22-18587R1Effectiveness, acceptability and feasibility of technology-enabled health interventions for adolescents living with HIV in low- and middle-income countries: a systematic review protocolPLOS ONE

Dear Dr. Crowley,

Thank you for submitting your manuscript to PLOS ONE. After careful consideration, we feel that it has merit but does not fully meet PLOS ONE’s publication criteria as it currently stands. Therefore, we invite you to submit a revised version of the manuscript that addresses the points raised during the review process.

We look forward to receiving your revised manuscript.

Kind regards,

Muhammad Shahzad Aslam, Ph.D.,M.Phil., Pharm-D

Academic Editor

PLOS ONE

Journal Requirements:

Reviewers' comments:

Reviewer's Responses to Questions

**Comments to the Author**

1. Does the manuscript provide a valid rationale for the proposed study, with clearly identified and justified research questions?

Reviewer #1: Yes

Reviewer #2: Yes

2. Is the protocol technically sound and planned in a manner that will lead to a meaningful outcome and allow testing the stated hypotheses?

Reviewer #1: Yes

Reviewer #2: Partly

3. Is the methodology feasible and described in sufficient detail to allow the work to be replicable?

Reviewer #1: Yes

Reviewer #2: Yes

4. Have the authors described where all data underlying the findings will be made available when the study is complete?

Reviewer #1: No

Reviewer #2: Yes

5. Is the manuscript presented in an intelligible fashion and written in standard English?

Reviewer #1: Yes

Reviewer #2: Yes

6. Review Comments to the Author

You may also provide optional suggestions and comments to authors that they might find helpful in planning their study.

Reviewer #1: Dear authors,

This is a very well-written protocol study, and now all my concerns are fulfilled.

Thanks for addressing all comments.

Reviewer #2: Dear Author,

Thank you for addressing all my comments. Well done as the manuscript has improved.

I still have one minor issue regarding the first two sentences of paragraph line 278 when you report your method for qualitative studies as I had troubles understanding them at first:

“Data on acceptability, feasibility, usability/functionality and perceived usefulness will be synthesised and reported narratively. Thematic qualitative analysis will be used to extract codes, categories and themes from the data related to acceptability, feasibility, usability/functionality and perceived usefulness.”

I suggest changing this in this way, if I am correct and if it does apply to what you intend to do:

“Data on acceptability, feasibility, usability/functionality and perceived usefulness from qualitative studies will be explored using thematic analysis and reported narratively.

Good luck with the review and its publication.

7. PLOS authors have the option to publish the peer review history of their article (what does this mean?). If published, this will include your full peer review and any attached files.

Reviewer #1: No

Reviewer #2: No

---

## [Author Response · Author response to Decision Letter 1]

31 Jan 2023

We checked the reference list and added DOI numbers for all the studies that were applicable. There were no retracted articles.

Reviewer 1 - no comments

Reviewer 1 - we have accepted the suggestion of the reviewer to revise the sentence.

---

## [Decision Letter · Decision Letter 2]

3 Feb 2023

Effectiveness, acceptability and feasibility of technology-enabled health interventions for adolescents living with HIV in low- and middle-income countries: a systematic review protocol

PONE-D-22-18587R2

Dear,

We’re pleased to inform you that your manuscript has been judged scientifically suitable for publication and will be formally accepted for publication once it meets all outstanding technical requirements.

Kind regards,

Muhammad Shahzad Aslam, Ph.D.,M.Phil., Pharm-D

Academic Editor

PLOS ONE

Additional Editor Comments (optional):

Reviewers' comments:

Reviewer's Responses to Questions

**Comments to the Author**

1. Does the manuscript provide a valid rationale for the proposed study, with clearly identified and justified research questions?

Reviewer #1: Yes

Reviewer #2: Yes

2. Is the protocol technically sound and planned in a manner that will lead to a meaningful outcome and allow testing the stated hypotheses?

Reviewer #1: Yes

Reviewer #2: Yes

3. Is the methodology feasible and described in sufficient detail to allow the work to be replicable?

Reviewer #1: Yes

Reviewer #2: Yes

4. Have the authors described where all data underlying the findings will be made available when the study is complete?

Reviewer #1: Yes

Reviewer #2: Yes

5. Is the manuscript presented in an intelligible fashion and written in standard English?

Reviewer #1: Yes

Reviewer #2: Yes

6. Review Comments to the Author

You may also provide optional suggestions and comments to authors that they might find helpful in planning their study.

Reviewer #1: Dear authors,

This is a very well-written protocol study,

and now all my concerns are fulfilled.

Thanks for addressing all comment

Reviewer #2: Dear author, thank you for your response. I have no further comments. Good luck for the review publication.

7. PLOS authors have the option to publish the peer review history of their article (what does this mean?). If published, this will include your full peer review and any attached files.

Reviewer #1: No

Reviewer #2: No

---

## [Editor Report · Acceptance letter]

7 Feb 2023

PONE-D-22-18587R2 

Effectiveness, acceptability, and feasibility of technology-enabled health interventions for adolescents living with HIV in low- and middle-income countries: a systematic review protocol 

Dear Dr. Crowley:

I'm pleased to inform you that your manuscript has been deemed suitable for publication in PLOS ONE. Congratulations! Your manuscript is now with our production department. 

Kind regards, 

on behalf of

Dr. Muhammad Shahzad Aslam 

Academic Editor

PLOS ONE